# Learning of Patch-Based Smooth-Plus-Sparse Models for Image Reconstruction

Stanislas Ducotterd[1], Sebastian Neumayer[2], Michael Unser[1]

[1]École polytechnique fédérale de Lausanne, [2] Technische Universität Chemnitz

stanislas.ducotterd@epfl.ch,
sebastian.neumayer@mathematik.tu-chemnitz.de,
michael.unser@epfl.ch

We aim at the solution of inverse problems in imaging, by combining a penalized sparse representation of image patches with an unconstrained smooth one. This allows for a straightforward interpretation of the reconstruction. We formulate the optimization as a bilevel problem. The inner problem deploys classical algorithms while the outer problem optimizes the dictionary and the regularizer parameters through supervised learning. The process is carried out via implicit differentiation and gradient-based optimization. We evaluate our method for denoising, super-resolution, and compressed-sensing magnetic-resonance imaging. We compare it to other classical models as well as deep-learning-based methods and show that it always outperforms the former and also the latter in some instances.

## 1. Introduction

We aim to learn a smooth-plus-sparse model for the resolution of linear inverse problems [1]. More precisely, given a measurement operator $\mathbf{H} \in \mathbb{R}^{m \times n}$ and noisy measurements $\mathbf{y} \in \mathbb{R}^m$, we want to find the underlying ground truth $\mathbf{x}^* \in \mathbb{R}^n$ that satisfies $\mathbf{H}\mathbf{x}^* \approx \mathbf{y}$. As $\mathbf{H}$ is commonly ill-conditioned or even singular, direct inversion fails and variational regularization can be deployed instead, as in

$$\underset{\mathbf{x} \in \mathbb{R}^n}{\arg\min} \frac{1}{2}\|\mathbf{H}\mathbf{x} - \mathbf{y}\|^2 + \mathcal{R}(\mathbf{x}). \tag{1}$$

Here, the first term ensures data consistency, and the regularizer $\mathcal{R} \colon \mathbb{R}^n \to \mathbb{R}_{\geq 0}$ encodes prior information. We deploy the patch-based regularizer $\mathcal{R}(\mathbf{x}) = \sum_{k=1}^n \mathcal{R}_k(\mathbf{P}_k \mathbf{x})$, where $\mathbf{P}_k \in \mathbb{R}^{d \times n}$ extracts a patch of size $\sqrt{d} \times \sqrt{d}$ at pixel $k$. To each patch, we apply dictionary-based regularizers $\mathcal{R}_k \colon \mathbb{R}^d \to \mathbb{R}_{\geq 0}$ of the form

$$\mathcal{R}_k(\mathbf{P}_k \mathbf{x}) = \min_{\boldsymbol{\alpha}_k \in \mathbb{R}^p} \frac{\beta}{2}\|\mathbf{P}_k \mathbf{x} - \mathbf{D}\boldsymbol{\alpha}_k\|^2 + \lambda R(\boldsymbol{\alpha}_k) \tag{2}$$

with weights $\beta, \lambda > 0$, a synthesis dictionary $\mathbf{D} \in \mathbb{R}^{d \times p}$, coefficients $\boldsymbol{\alpha}_k \in \mathbb{R}^p$, and a sparsity prior $R \colon \mathbb{R}^p \to \mathbb{R}_{\geq 0}$. Essentially, (2) enforces that each patch has an $R$-sparse (low regularization cost) representation in the dictionary $\mathbf{D}$. These considerations lead to the patch-based reconstruction

$$\mathbf{x}^* \in \underset{\mathbf{x} \in \mathbb{R}^n}{\arg\min} \min_{(\boldsymbol{\alpha}_k)_{k=1}^n} \frac{1}{2}\|\mathbf{H}\mathbf{x} - \mathbf{y}\|^2 + \sum_{k=1}^n \frac{\beta}{2}\|\mathbf{P}_k \mathbf{x} - \mathbf{D}\boldsymbol{\alpha}_k\|^2 + \lambda R(\boldsymbol{\alpha}_k), \tag{3}$$

where we consider $\mathbf{D}$ and $R$ as learnable parameters. In particular, they will be chosen such that $\mathbf{x}^*$ is a high-quality reconstruction. Such data-driven variational models for solving inverse problems have become increasingly popular in recent years [2, 3].

Given paired training data $(\mathbf{x}_m, \mathbf{y}_m)_{m=1}^M$, we use supervised learning to obtain the model parameters $\mathbf{D}$ and $R$ such that the solutions $\mathbf{x}_k^*$ of (3) minimize the reconstruction error $\mathcal{L}$ on the training set given by

$$\mathcal{L}(\mathbf{D}, R) = \frac{1}{M} \sum_{m=1}^M \|\mathbf{x}_m^* - \mathbf{x}_m\|_1. \tag{4}$$

Second Conference on Parsimony and Learning (CPAL 2025).

For this, we need to solve a bilevel task with two subproblems:

- the inner problem in (3), namely, a search for the optimal $\mathbf{x}^*$ with classical optimization algorithms such as inertial proximal alternating linearized minimization (iPALM) [4];
- the outer problem (4), where we learn $\mathbf{D}$ and $R$. Gradient-based algorithms such as ADAM [5] require the derivatives of $\mathbf{x}_m^*$ with respect to the parameters. These are accessible via implicit differentiation, popularized with the deep equilibrium (DEQ) framework [6].

**Contribution** We propose a learning scheme to determine the parameters in (3). To improve image reconstruction, we found it helpful to ignore low-frequency components in the regularization (2) of each patch $\mathbf{P}_k\mathbf{x}$. Hence, we modify $\mathbf{P}_k$ into $\hat{\mathbf{P}}_k = (\mathbf{I} - \mathbf{Q}\mathbf{Q}^T)\mathbf{P}_k$ with a learnable analysis dictionary $\mathbf{Q} \in \mathbb{R}^{d \times p_2}$. Since $(\mathbf{I} - \mathbf{Q}\mathbf{Q}^T)$ projects onto $\ker(\mathbf{Q}^T)$, this means that we remove a learned subspace of $\mathbb{R}^d$ during the patch extraction. This leads to a decomposition of the reconstruction $\mathbf{x}^*$ into a smooth component induced by $\mathbf{Q}$, and a sparse one induced by $\mathbf{D}$. For the optimization in (3), we show that all linear operators are expressible as convolutions. Hence, the minimization in (3) amounts to a search for the fixed point of a two-layer convolutional neural network without any explicit extraction of patches. After having trained the dictionaries $\mathbf{D}$ and $\mathbf{Q}$, and the regularizer $R$, we evaluate the model on denoising, super-resolution and compressed-sensing magnetic-resonance imaging (CS-MRI). We compare our results with similar classical methods such as TV [7], K-SVD [8] and BM3D [9] as well as two deep-learning-based methods, namely, DRUNet [10] and Prox-DRUNet [11]. Our code is accessible on Github[1].

## 2. Related Work

Patch-based techniques have a long history [12–15] in image reconstruction. Since the empirical patch distributions are similar at different image scales, they are well-suited for image characterization [16]. In principle, we can interpret our model (3) as a special case of the generic expected patch log likelihood model [17] with a dictionary-based sparsity prior [18]. To minimize this objective, one can deploy half quadratic splitting, which makes the surrogate objective in each step similar to (3). Commonly deployed priors for the coefficients $\boldsymbol{\alpha}_k$ are $\ell_0, \ell_1$, or non-convex relaxations between these two. However, the $\beta$ is successively increased during the iterations. Over the years, patch-based modeling ideas have also been incorporated into data-driven approaches.

**Dictionary Learning** Dictionary-based priors are well-established [13, 19]. Aside from computing the $\boldsymbol{\alpha}$, most approaches also adapt $\mathbf{D}$ during the reconstruction process [20, 21], which we only do during the training. To perform the updates, the most popular approach used to be alternating minimization with updates of $\mathbf{D}$ based on either the SVD, [8], sequential schemes [20], or the proximal gradient method [22]. Alternatives are based on block coordinate descent [21, 23]. Nowadays, after the pioneering work of [24], algorithm unrolling is commonly used to optimize over $\mathbf{D}$ [25, 26]. We pursue a different strategy for training, namely, a bilevel approach with implicit differentiation to compute the required gradient $\nabla_{\mathbf{D}}\mathbf{x}_k$, see also [27, 28].

**Convolutional Dictionaries** There is a line of work that uses convolutional dictionaries to improve the computational efficiency [29–31]. A recent extension aims to also incorporate multiscale modeling [32]. In contrast to our model (3), $\mathbf{x}$ is modeled as the convolution of a dictionary $\mathbf{D}$ with sparse coefficients $\boldsymbol{\alpha}$. As it leads to more degrees of freedom, the approach might be suboptimal for challenging inverse problems [21].

**Variational Image Decomposition** Images can be often modeled as the superposition of cartoon parts (piecewise smooth) and texture parts (high-frequency content) [33]. With synthesis-based variational models, one can perform such a decomposition based on morphological components [34], specific texture priors [35, 36], specific dictionaries [37], or generative models [38]. In contrast to all these models, our cartoon part is actually smooth and does not contain discontinuities. Hence,

---

[1]https://github.com/StanislasDucotterd/Smooth-Plus-Sparse-Model

we do not have the issue that there is an ambiguity between the two components. Closest in spirit to our work is [39], where the authors also aim to learn two different dictionaries. However, they impose fewer strict constraints on the dictionaries and treat each patch separately. Hence, their approach does not readily generalizes to inverse problems.

## 3. Method

We seek to learn two dictionaries of atoms $\mathbf{D} \in \mathbb{R}^{d \times p_1}$ and $\mathbf{Q} \in \mathbb{R}^{d \times p_2}$ with $\mathbf{Q}^T \mathbf{D} = \mathbf{0}$ and $\mathbf{Q}^T \mathbf{Q} = \mathbf{I}$, as well as a regularizer $R$ such that any minimizer $(\mathbf{x}^*, \boldsymbol{\alpha}^*) \subset \mathbb{R}^n \times (\mathbb{R}^{p_1})^n$ of the objective

$$J_{\mathbf{D},\mathbf{Q},R,\mathbf{y}}(\mathbf{x}, \boldsymbol{\alpha}) = \frac{1}{2}\|\mathbf{H}\mathbf{x} - \mathbf{y}\|^2 + \min_{(\mathbf{c}_k)_{k=1}^n}\left(\sum_{k=1}^n \frac{\beta}{2}\|\mathbf{P}_k\mathbf{x} - \mathbf{Q}\mathbf{c}_k - \mathbf{D}\boldsymbol{\alpha}_k\|^2 + \lambda R(\boldsymbol{\alpha}_k)\right) \tag{5}$$

corresponds to a high-quality reconstruction $\mathbf{x}^*$ of the data $\mathbf{y}$. Here, the constraint $\mathbf{Q}^T \mathbf{D} = \mathbf{0}$ ensures that the representations $\mathbf{D}\boldsymbol{\alpha}_k$ and $\mathbf{Q}\mathbf{c}_k$ are contained in two orthogonal subspaces. Since only the column space of $\mathbf{Q}$ matters in (5), the constraint that $\mathbf{Q}$ be a tight frame is not restrictive. Then, $(\mathbf{I} - \mathbf{Q}\mathbf{Q}^T)$ is the orthogonal projection onto the subspace $\ker(\mathbf{Q}^T)$, which allows us to rewrite (5) as

$$J_{\mathbf{D},\mathbf{Q},R,\mathbf{y}}(\mathbf{x}, \boldsymbol{\alpha}) = \frac{1}{2}\|\mathbf{H}\mathbf{x} - \mathbf{y}\|^2 + \sum_{k=1}^n \frac{\beta}{2}\|\hat{\mathbf{P}}_k\mathbf{x} - \mathbf{D}\boldsymbol{\alpha}_k\|^2 + \lambda R(\boldsymbol{\alpha}_k), \tag{6}$$

where $\hat{\mathbf{P}}_k = (\mathbf{I} - \mathbf{Q}\mathbf{Q}^T)\mathbf{P}_k$. By replacing $\mathbf{P}_k$ with $\hat{\mathbf{P}}_k$ in (3), we follow the paradigm that not everything in a patch $\mathbf{P}_k\mathbf{x}$ should be penalized. In particular, it makes sense to relax constraints on the mean (see also [28]) and on some low-frequency components. Due to the constraint $\mathbf{Q}^T \mathbf{D} = \mathbf{0}$, we can rewrite our generalized patch regularizer (2) as

$$\mathcal{R}_k(\mathbf{x}) = \min_{\mathbf{u} \in \mathbb{R}^d}\left(\frac{\beta}{2}\|(\mathbf{I} - \mathbf{Q}\mathbf{Q}^T)(\mathbf{x} - \mathbf{u})\|^2 + \min_{\boldsymbol{\alpha} \text{ s.t. } \mathbf{D}\boldsymbol{\alpha} = \mathbf{u}} \lambda R(\boldsymbol{\alpha})\right), \tag{7}$$

namely, as the infimal convolution of an analysis- and a synthesis-based regularizer. To evaluate the bilevel training objective (4) with the additional parameter $\mathbf{Q}$, we need to minimize the objective (6). This is detailed in the Section 3.1.

### 3.1. Training of the Model—Inner Optimization

We minimize the objective (6) using the iPALM algorithm [4], which consists of the updates

$$\boldsymbol{\beta}_k^{(m)} = \boldsymbol{\alpha}_k^{(m)} + \frac{m-1}{m+2}\left(\boldsymbol{\alpha}_k^{(m)} - \boldsymbol{\alpha}_k^{(m-1)}\right) \tag{8}$$

$$\boldsymbol{\alpha}_k^{(m+1)} = \text{Prox}_{\gamma_1 \lambda R}\left(\boldsymbol{\beta}_k^{(m)} - \gamma_1 \mathbf{D}^T\left(\mathbf{D}\boldsymbol{\beta}_k^{(m)} - \hat{\mathbf{P}}_k\mathbf{x}^{(m)}\right)\right) \tag{9}$$

$$\mathbf{z}^{(m)} = \mathbf{x}^{(m)} + \frac{m-1}{m+2}\left(\mathbf{x}^{(m)} - \mathbf{x}^{(m-1)}\right) \tag{10}$$

$$\mathbf{x}^{(m+1)} = \mathbf{z}^{(m)} - \gamma_2\left(\mathbf{H}^T(\mathbf{H}\mathbf{z}^{(m)} - \mathbf{y}) + \sum_{k=1}^n \beta\hat{\mathbf{P}}_k^T\left(\hat{\mathbf{P}}_k\mathbf{z}^{(m)} - \mathbf{D}\boldsymbol{\alpha}_k^{(m+1)}\right)\right), \tag{11}$$

where $\gamma_1 = 0.99/\|\mathbf{D}^T\mathbf{D}\|$ and $\gamma_2 = 0.99/\|\beta \sum_{k=1}^n \hat{\mathbf{P}}_k^T\hat{\mathbf{P}}_k\|$. The convergence of iPALM under some weak conditions is guaranteed by [4, Thm. 4.1].

In Iterations (8)–(11), an important detail is the handling of boundary for the patch extraction $\mathbf{P}_k\mathbf{x}$. Specifically, for an image with $n$ pixels, we extract $n$ patches by applying circular padding to the image. This extension allows us to express all the relevant linear operators as convolutions. For this, we represent the code $\boldsymbol{\alpha}$ as a $p_1$-channel image with the same spatial dimensions as $\mathbf{x}$. Then, $\boldsymbol{\alpha}_k \in \mathbb{R}^{p_1}$ refers to a single pixel on that code. The matrix $\mathbf{D}^T\mathbf{D} \in \mathbb{R}^{p_1 \times p_1}$ in (9) is applied to every pixel on the code using a 1×1 convolution. The operator $\sum_{k=1}^n \hat{\mathbf{P}}_k^T\mathbf{D}$ in (11) maps a $p_1$-channel image to a single-channel one and can be implemented as a $\sqrt{d} \times \sqrt{d}$ convolution whose filters

**Algorithm 1:** Parameterization of $\mathcal{B}$

---

**Input :** $\tilde{\mathbf{D}} \in \mathbb{R}^{d \times p_1}, \tilde{\mathbf{Q}} \in \mathbb{R}^{d \times (p_2 - 1)}$
**Output :** $\mathbf{D}, \mathbf{Q} \in \mathcal{B}$
Remove the mean of each column of $\tilde{\mathbf{Q}}$
$\mathbf{Q} = \text{Björck}(\tilde{\mathbf{Q}})$
Add a non-zero constant column to $\mathbf{Q}$
$\mathbf{D} = (\mathbf{I} - \mathbf{Q}\mathbf{Q}^T)\tilde{\mathbf{D}}$
$\mathbf{D} \leftarrow \mathbf{D} \, \mathbf{diag}((\|\mathbf{d}_k\|^{-1})_{k=1}^{p_1})$
Divide $\mathbf{D}$ by its spectral norm
**return** $\mathbf{D}, \mathbf{Q}$

---

are the flipped atoms of $\mathbf{D}$. Since $\sum_{k=1}^{n} \mathbf{D}^T \hat{\mathbf{P}}_k$ is the transpose of $\sum_{k=1}^{n} \hat{\mathbf{P}}_k^T \mathbf{D}$, we can similarly implement it as a convolution. Thus, we can evaluate (9) for each pixel using two convolutions and one nonlinearity.

For $\sum_{k=1}^{n} \hat{\mathbf{P}}_k^T \hat{\mathbf{P}}_k$ in (11), we now prove that it is Linear Shift-Invariant and, hence, a convolution. Therefore, we can also efficiently compute $\gamma_2$ in (11) with the Fourier transform.

**Lemma 1.** *The operator $\sum_{k=1}^{n} \hat{\mathbf{P}}_k^T \hat{\mathbf{P}}_k$ is Linear Shift-Invariant.*

*Proof.* We get that $\hat{\mathbf{P}}_k = (\mathbf{I} - \mathbf{Q}\mathbf{Q}^T)\mathbf{P}_k$ is linear since both $(\mathbf{I} - \mathbf{Q}\mathbf{Q}^T)$ and $\mathbf{P}_k$ are linear operators. Hence, the same holds for its transpose and the sum $\sum_{k=1}^{n} \hat{\mathbf{P}}_k^T \hat{\mathbf{P}}_k$. For the need of the proof, we express the sum over the two-dimensional indices of the image $\mathbf{k} \in \Omega = [1...H] \times [1...W]$, where $H, W \in \mathbb{Z}$ are the height and width of the image. We get that for any $\mathbf{m} \in \mathbb{Z}^2$ with associated shift operator $T_{\mathbf{m}}$, it holds that

$$\sum_{\mathbf{k} \in \Omega} \hat{\mathbf{P}}_{\mathbf{k}}^T \hat{\mathbf{P}}_{\mathbf{k}} T_{\mathbf{m}} \mathbf{x} = \sum_{\mathbf{k} \in \Omega} \hat{\mathbf{P}}_{\mathbf{k}}^T \hat{\mathbf{P}}_{(\mathbf{k}-\mathbf{m})^*} \mathbf{x} = T_{\mathbf{m}} \sum_{\mathbf{k} \in \Omega} \hat{\mathbf{P}}_{(\mathbf{k}-\mathbf{m})^*}^T \hat{\mathbf{P}}_{(\mathbf{k}-\mathbf{m})^*} \mathbf{x} = T_{\mathbf{m}} \sum_{\mathbf{k} \in \Omega} \hat{\mathbf{P}}_{\mathbf{k}}^T \hat{\mathbf{P}}_{\mathbf{k}} \mathbf{x}, \qquad (12)$$

where all equalities rely on the circular padding and where $\mathbf{p}^* = \mathbf{p} \bmod (H + 1, W + 1)$. $\qquad \square$

Therefore, solving (6) amounts to finding the fixed point of a two-layer convolutional network.

## 3.2. Training of the Model—Outer Optimization

All the parameters in (6) are learned through implicit differentiation with the `torchdeq` library [40]. In the following, we provide the parameterization details.

**Dictionaries** We enforced the constraints $\mathbf{Q}^T \mathbf{Q} = \mathbf{I}$ and $\mathbf{Q}^T \mathbf{D} = \mathbf{0}$ in Section 3.1 to ensure the equivalence of (5) and (6). Now, we impose additional constraints on the dictionaries $\mathbf{D}$ and $\mathbf{Q}$. More precisely, if we denote the $k$th column of $\mathbf{D}$ by $\mathbf{d}_k$, which corresponds to the $k$th atom, the feasible set reads

$$\mathcal{B} = \left\{ \mathbf{D} \in \mathbb{R}^{d \times p_1}, \mathbf{Q} \in \mathbb{R}^{d \times p_2} : \|\mathbf{D}\|_2 = 1, \|\mathbf{d}_k\| = \|\mathbf{d}_1\| \, \forall k, \mathbf{Q}^T \mathbf{Q} = \mathbf{I}, \mathbf{Q}^T \mathbf{D} = \mathbf{0} \right\}. \qquad (13)$$

Due to the normalization $\|\mathbf{D}\|_2 = 1$, we get that $\gamma_1 = 0.99$ in (9) for the inner optimization. The norm constraint $\|\mathbf{d}_k\| = \|\mathbf{d}_1\|, 1 \le k \le p_1$, ensures that the relative importance of each atom $\mathbf{d}_k$ gets encoded in $R$ and not in $\mathbf{D}$ itself. This makes the objective (6) more interpretable. In Algorithm 1, we explicitly enforce that $\mathbf{Q}$ contains a constant atom. For an efficient training, it is important to embed the constraints into the forward pass, through a suitable parameterization.

We parameterize the elements $\{\mathbf{D}, \mathbf{Q}\}$ in $\mathcal{B}$ by unconstrained matrices $\tilde{\mathbf{D}} \in \mathbb{R}^{d \times p_1}, \tilde{\mathbf{Q}} \in \mathbb{R}^{d \times (p_2-1)}$, as outlined in Algorithm 1. There, the Björck algorithm (of order $p = 1$) is used to parameterize the element $\mathbf{Q}$ with $\mathbf{Q}^T \mathbf{Q} = \mathbf{I}$ [41]. It sets $\mathbf{Q}_0 = \tilde{\mathbf{Q}}$ and performs the fixed-point iteration

$$\mathbf{Q}_{k+1} = \frac{1}{2} \mathbf{Q}_k (3 \cdot \mathbf{I} - \mathbf{Q}_k^T \mathbf{Q}_k). \qquad (14)$$

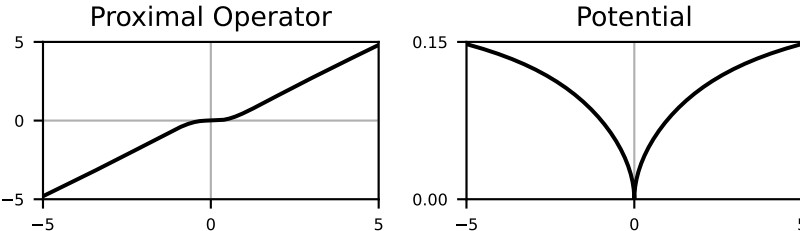

Figure 1: The proximal operator of the non-convex model and the corresponding potential computed numerically when $\gamma = 2$ and $\tau = 1$.

If we remove the mean of each column in $\mathbf{Q}_0 = \tilde{\mathbf{Q}}$, we can recursively verify that it holds that $\mathbf{1}_d^T \mathbf{Q}_{k+1} = \frac{1}{2}\mathbf{1}_d^T \mathbf{Q}_k(3\cdot\mathbf{I} - \mathbf{Q}_k^T\mathbf{A}_k) = \mathbf{0}_{p_1}^T$ for the iterations (14). Hence, the Björck algorithm preserves the zero mean. In practice, 15 iterations of (14) are enough to obtain $\mathbf{Q}$ with $\mathbf{Q}^T\mathbf{Q} \approx \mathbf{I}$.

**Regularizers** We deploy both convex and non-convex regularizers $R$ in (3). As convex regularizer, we choose the mixed group sparsity $(\ell_1 - \ell_2)$-norm with learnable non-negative weights $\tau = \{\tau_l\}_{l=1}^{p_1}$, as given by

$$R_\tau(\boldsymbol{\alpha}) = \sum_{k=1}^{n/4} \sum_{p=1}^{p_1} \tau_l \left(\alpha_{4k,p}^2 + \alpha_{4k-1,p}^2 + \alpha_{4k-2,p}^2 + \alpha_{4k-3,p}^2\right)^{1/2}. \tag{15}$$

We refer to the resulting model as a convex patch-based regularizer (CPR).

For the non-convex case, we (implicitly) choose a pixel-wise regularizer $R$ whose proximal operator with learnable non-negative weights $\{\tau_l\}_{p=1}^{p_1}$ is expressed as

$$\mathrm{Prox}_R(\boldsymbol{\alpha}) = ((\varphi_p(\alpha_{k,p}))_{k=1}^n)_{p=1}^{p_1}, \quad \text{where } \varphi_p(x) = \frac{x|x|^\gamma}{\tau_p^\gamma + |x|^\gamma}, \gamma > 0. \tag{16}$$

Note that (16) converges pointwise to the hard threshold $x \mapsto x \cdot \mathbf{1}_{\mathbb{R}\setminus[-\tau,\tau]}(x)$ as $\gamma \to \infty$. A similar approximation is also used in [42]. For our experiments, we use $\gamma \leq 2$, which remains far from the hard-threshold function. We display a numerical approximation of the associated $R$ in Figure 1, and refer to the associated model as the non-convex patch-based regularizer (NCPR).

## 4. Experiments

### 4.1. Training on Denoising

We learn the parameters $\{\mathbf{D}, \mathbf{Q}, R\}$ in (6) on a basic denoising task. The training dataset consists of 238400 small images of size $(40 \times 40)$ with values in $[0, 1]$ from the BSD500 images [43]. For the experiment, we corrupt them with Gaussian noise with $\sigma \in \{5/255, 25/255\}$. We use atoms of size $13 \times 13$, 120 for $\mathbf{Q}$ and 200 for $\mathbf{D}$. The iPALM algorithm terminates if the relative difference of the iterates is smaller than $10^{-4}$. For the implicit differentiation of the inner solutions $\mathbf{x}_m^*$, we use 75 iterations of the Anderson algorithm [44] for $\sigma = 5/255$ and 50 iterations of the Broyden algorithm [45] for $\sigma = 25/255$. We use the ADAM optimizer with a batch size of 16, a learning rate of $2 \cdot 10^{-4}$ for $\{\mathbf{D}, \mathbf{Q}\}$ and a learning rate of $10^{-3}$ for $R$ and $\beta$. We train for two epochs and decay the learning rate 10 times per epoch by 0.75 and 0.9 for the CPR and NCPR models, respectively. The whole training takes approximately 10 hours on a Tesla V100 GPU.

We report the metrics on the BSD68 dataset in Table 1, and compare our methods with

- the convex TV regularization [7];
- the patch-based methods K-SVD [8] and BM3D [9];

| Free Atoms (**Q**) | Regularized Atoms (**D**) |
|---|---|

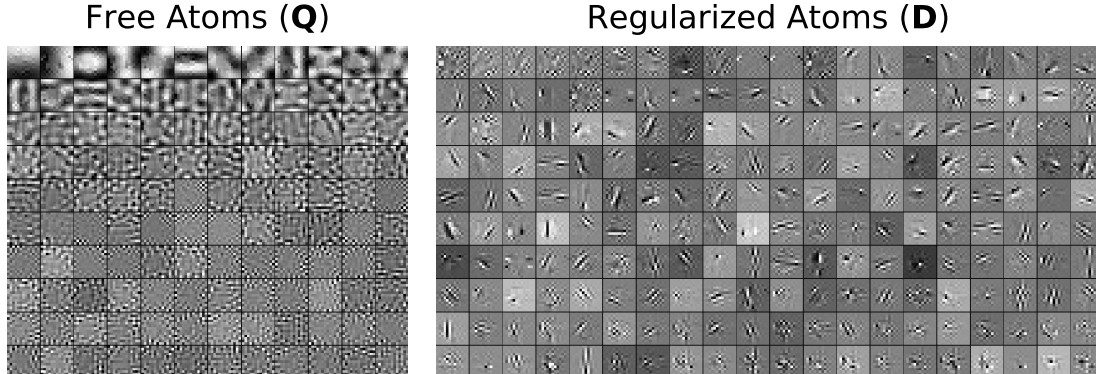

Figure 2: Learned atoms for the NCPR model with $\sigma = 25/255$.

- the deep learning methods DRUNet [10] and Prox-DRUNet [11].

The DRUNet is a deep denoiser that has around 32 million parameters, 600 times more than our method. It achieves state-of-the-art performance in denoising and many image-reconstruction tasks. The Prox-DRUNet constrains the parameters such that it is (approximately) the proximal operator of an unknown non-convex potential. We can see that both our models significantly outperform TV and our non-convex model outperforms K-SVD (which approximates the $\ell_0$-norm and is therefore also not convex) and BM3D (which does not result from the minimization of an objective).

## 4.2. Visualization of the Model

We show the learned free and regularized atoms (corresponding to **Q** and **D**, respectively) in Figure 2. Note that **Q** represents a $p_2$-dimensional subspace of $\mathbb{R}^d$. For any orthonormal matrix $\mathbf{R} \in \mathbb{R}^{p_2 \times p_2}$, the substitution of $\mathbf{QR}$ for **Q** encodes the same model. For visualization purposes, we computed **R** such that the first atom (top left) and last atom (bottom right) in the figure have the highest (lowest, respectively) variance in all of the overlapping patches of the BSD68 dataset. Note that **Q** also has an atom enforced to be constant which is not displayed here.

The regularized atoms are sorted by the value of their respective $\tau_p$, so that, the atom on the top left is the least penalized and the atom on the bottom right is the most penalized in the model. We show a similar plot for the other noise level and the other regularizer in Section A.1. We can split any reconstruction $\mathbf{x}^*$ into two components based on

$$\left( \mathbf{H}^T\mathbf{H} + \beta \sum_{k=1}^{n} \mathbf{P}_k^T(\mathbf{I} - \mathbf{QQ}^T)\mathbf{P}_k \right)\mathbf{x}^* = \mathbf{H}^T\mathbf{y} + \beta \sum_{k=1}^{n} \mathbf{P}_k^T\mathbf{D}\boldsymbol{\alpha}_k^*, \tag{17}$$

namely, a smooth part corresponding to a generalized Tikhonov regularization, and a sparse one induced by the reconstruction from **D**. Visual examples are given in Figure 3 and Sections A.1,A.2.

## 4.3. Inverse Problems

We benchmark CPR and NCPR for image super-resolution and CS-MRI. For this, we set **H** in (6) (which was the identity during training) to the linear operator that corresponds to the imaging modality. Again, we compare with the models in Section 4.1, except for the K-SVD and BM3D, which are not suited to inverse problems.

Image reconstruction with these methods (except DRUNet) requires the minimization of an objective function. We use the Chambolle algorithm [46] for TV, the iPALM algorithm for our models, and the Douglas-Ratchford splitting algorithm [47] for Prox-DRUNet. The optimization is termi-

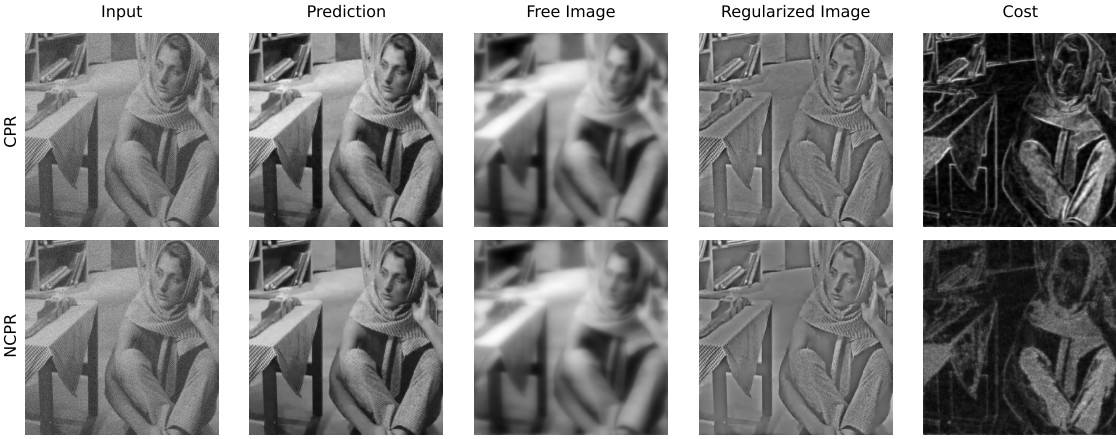

Figure 3: Denoising with $\sigma = 25/255$: decomposition of $\mathbf{x}^*$ into the free-atom and constrained-atom dictionaries. The last column shows the cost associated with each local patch $\mathbf{P}_k\mathbf{x}^*$.

nated when the relative difference of the iterates is below $10^{-5}$. For DRUNet, we simply follow the algorithm proposed in [10].

To deploy the models, we tune their hyperparameters on a validation set using the coarse-to-fine grid search from [48]. Then, the performance is reported on a dedicated test set. For TV, we only tune the regularization strength $\lambda$ in (1). For Prox-DRUNet, we tune $\lambda$ and the noise level $\sigma$ over which the model was trained. For CPR, we tune the $\beta$ and $\lambda$ in (6). For NCPR we tune $\beta$ and a constant multiplying all the $\{\tau_l\}_{p=1}^{p_1}$ described in (16). For DRUNet, we use the recommended algorithm [10, Sec. 4.2] and tune the $\sigma_1$ and $\lambda$. While we use the recommended 40 iterations for super-resolution, we increase this number to 80 iterations for CS-MRI as this significantly improves the performance.

DRUNet uses a fixed number of steps and does not have any convergence guarantees. As result, its reliability depends heavily on empirical performance, which questions its robustness. This is why we changed the number of steps for the CS-MRI experiment. Prox-DRUNet mitigates this problem by constraining its network to approximately be a proximal operator. While it demonstrates convergence in practice, it lacks provable guarantees. This limitation arises from the NP-hardness of the computation the Lipschitz constant of a deep network [49], a key factor to ensure the theoretical convergence of the method [11, Thm. 4.4]. In constrast, our method has provable convergence guarantees and offers an interpretable decomposition of the reconstruction with the free and regularized atoms.

**Super-Resolution** Here, we investigate the super-resolution of microstructures. The ground truth consists of 2D slices of size $600 \times 600$. They are extracted from a volume of size $2560 \times 2560 \times 2120$ acquired at the Swiss light-source beamline TOMCAT. We choose $\mathbf{H}$ as a convolution with a $16 \times 16$ Gaussian kernel with standard deviation of 2 and a stride 4. When simulating the data $\mathbf{y} = \mathbf{H}\mathbf{x} + \mathbf{n}$ from the ground truth, we add Gaussian noise with $\sigma_{\mathbf{n}} = 0.01$. For all methods, we tune the hyperparameters with a single validation image. The resulting test performances on 100 images are reported in Table 2. The results are shown in Figure 4 and the decomposition from CPR and NCPR are shown in Section A.2.

**Compressed-Sensing MRI** We now look at the CS-MRI recovery of an image $\mathbf{x} \in \mathbb{R}^n$ from noisy measurements $\mathbf{y} = \mathbf{M}\mathbf{F}\mathbf{x} + \mathbf{n} \in \mathbb{C}^m$, where $\mathbf{M}$ is a subsampling mask (identity matrix with some missing entries), $\mathbf{F}$ is the discrete Fourier transform, and $\mathbf{n}$ is a complex Gaussian noise with $\sigma_{\mathbf{n}} = 2 \cdot 10^{-3}$ for both the real and imaginary parts. We consider two different types of masks. Their subsampling rate is determined by the acceleration factor $M_{\text{acc}}$ with the number of columns kept in

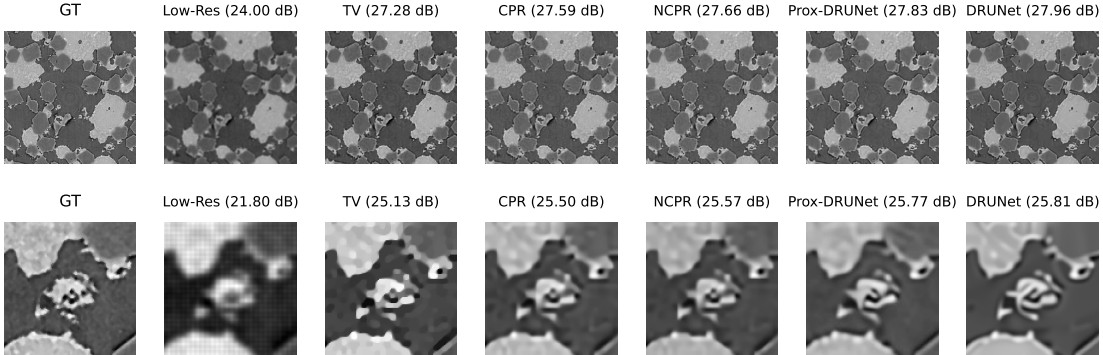

Figure 4: Reconstruction performances for the super-resolution task.

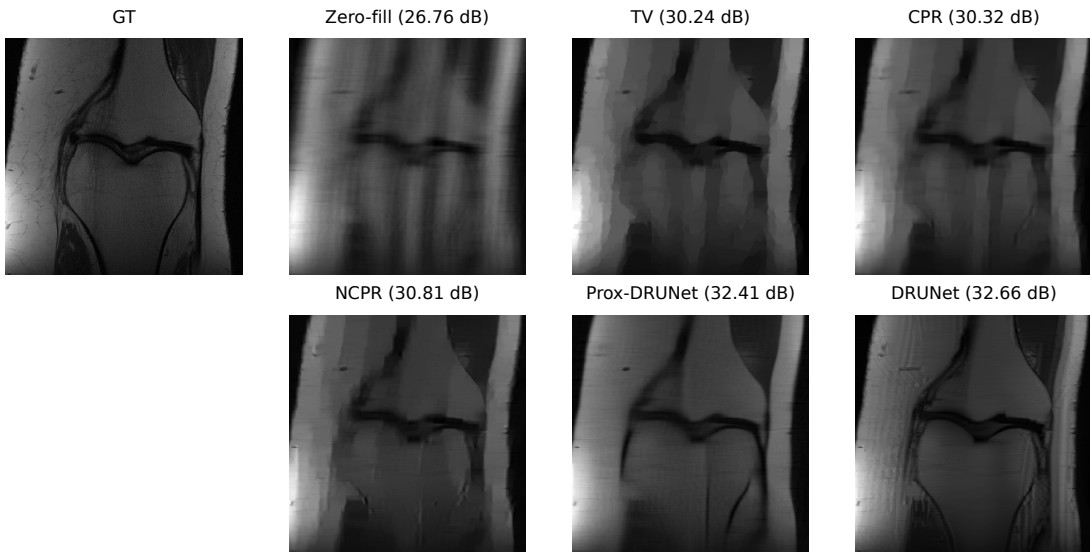

Figure 5: Reconstruction performances for 8-fold subsampling and PD data.

the k-space being proportional to $1/M_{\text{acc}}$. Our setups are 8-fold ($M_{\text{acc}} = 8$) and 16-fold ($M_{\text{acc}} = 16$). The ground truth comes from the fastMRI dataset [50] and we consider it with (PDFS) or without (PD) fat suppression. For each instance, we run the validation over 10 images and test on 50 other images.

In Table 3, we provide the PSNR values on centered ($320 \times 320$) patches. The deep-learning-based methods outperform the classical ones for the 8-fold subsampling and the 16-fold with PD. However, despite containing significantly fewer parameters and being trained on significantly fewer data, our NCPR model outperforms the deep-learning-based methods for 16-fold subsampling with PDFS data. We also provide visual results in Figures 5 and 6. While the deep-learning-based methods yield higher resolution in Figure 5, they still contain several artifacts. For example, we can see a black structure at the bottom of the Prox-DRUNet reconstruction. We can also see some oscillations on the left of the DRUNet reconstruction. In Figure 6, the deep-learning-based methods are outperformed by the NCPR. Note that several methods contain an almost-identical artifact on the bottom right of the image. We show the decomposition of both CPR and NCPR for those two examples in Section A.2. Suprisingly, due to the very low value of the optimal $\beta$ ($< 0.02$ for both), it seems that the free images are not smooth with the NCPR model. This makes sense as the noise is very low and the zero-filled signal contains valuable information which should not be smoothed by the model.

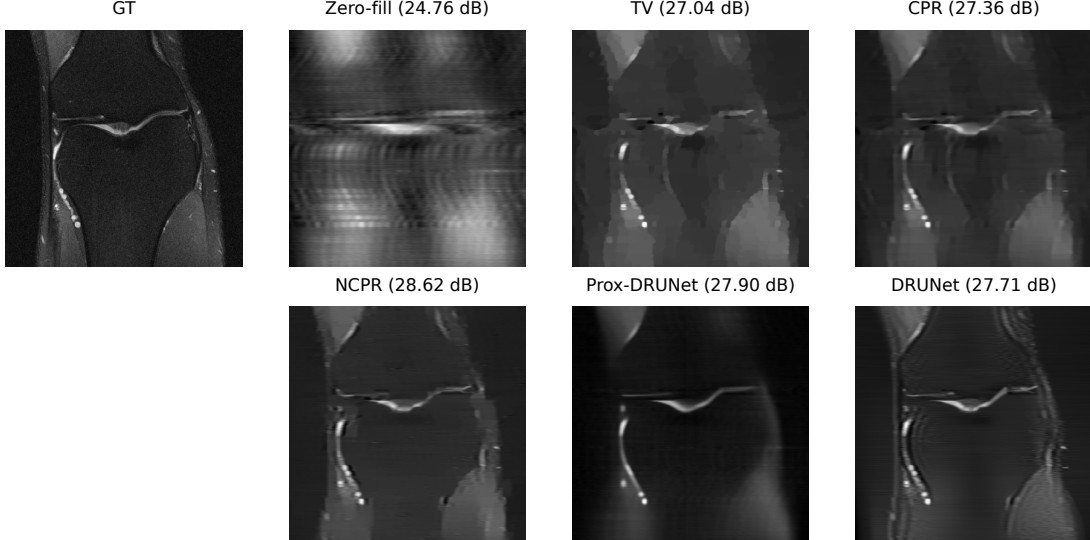

Figure 6: Reconstruction performances for 16-fold subsampling and PDFS data.

| Noise level | $\sigma=5/255$ | $\sigma=25/255$ |
|---|---|---|
| TV | 36.41 | 27.48 |
| K-SVD | - | 28.32 |
| BM3D | 37.54 | 28.60 |
| CPR (Ours) | 36.93 | 28.18 |
| NCPR (Ours) | 37.62 | 28.68 |
| Prox-DRUNet | 37.98 | 29.18 |
| DRUNet | **38.09** | **29.48** |

Table 1: PSNR values for the denoising of the BSD68 dataset.

| Super-resolution | PSNR | SSIM |
|---|---|---|
| Bicubic | 25.63 | 0.699 |
| TV | 27.69 | 0.763 |
| CPR (Ours) | 27.98 | 0.775 |
| NCPR (Ours) | 28.04 | 0.781 |
| Prox-DRUNet | 28.21 | **0.789** |
| DRUNet | **28.37** | 0.788 |

Table 2: Super-resolution on the SiC Diamonds dataset.

| | 8-fold | | | | 16-fold | | | |
|---|---|---|---|---|---|---|---|---|
| | PSNR | | SSIM | | PSNR | | SSIM | |
| | PD | PDFS | PD | PDFS | PD | PDFS | PD | PDFS |
| Zero-fill | 23.46 | 26.84 | 0.592 | 0.634 | 20.76 | 24.67 | 0.548 | 0.590 |
| TV | 25.76 | 28.70 | 0.666 | 0.654 | 21.78 | 25.71 | 0.576 | 0.598 |
| CPR | 25.82 | 28.77 | 0.672 | 0.667 | 21.81 | 25.68 | 0.583 | 0.602 |
| NCPR | 26.92 | 29.53 | 0.712 | 0.690 | 22.10 | **26.16** | 0.601 | **0.619** |
| Prox-DRUNet | 28.80 | 30.07 | 0.750 | **0.699** | 22.70 | 25.85 | **0.619** | 0.608 |
| DRUNet | **28.85** | **30.32** | **0.752** | **0.699** | **22.73** | 25.83 | 0.606 | 0.610 |

Table 3: PSNR values for MRI reconstruction on the fastMRI dataset.

# 5. Conclusion

In this paper, we have introduced a patch-based smooth-plus-sparse model for image reconstruction. Our approach integrates both synthesis and analysis dictionaries to facilitate a smooth reconstruction process. In particular, low-frequency components are allowed greater flexibility with the analysis prior, while high-frequency details are regularized through the use of the synthesis dictionary. The expression of our optimization process as a two-layer convolutional neural network allows us to train the synthesis dictionary and analysis prior in an efficient and effective manner. Experimental results show that our model can outperform state-of-the-art methods when the inverse problem involves very few measurements. This suggests a robustness stemming from our model principled design, which is less reliant on large datasets compared to deep-learning approaches.

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

# A. Appendix

## A.1. Additional Model Visualizations

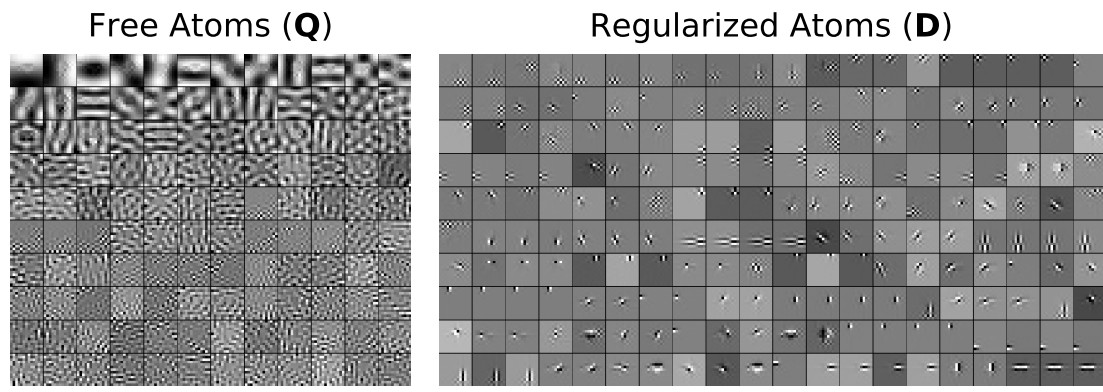

Figure 7: Learned atoms for the CPR model with $\sigma = 5/255$.

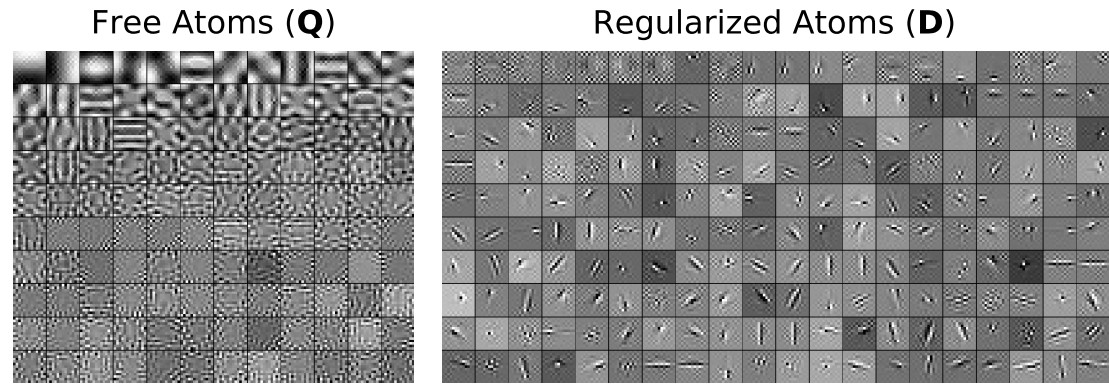

Figure 8: Learned atoms for the CPR model with $\sigma = 25/255$.

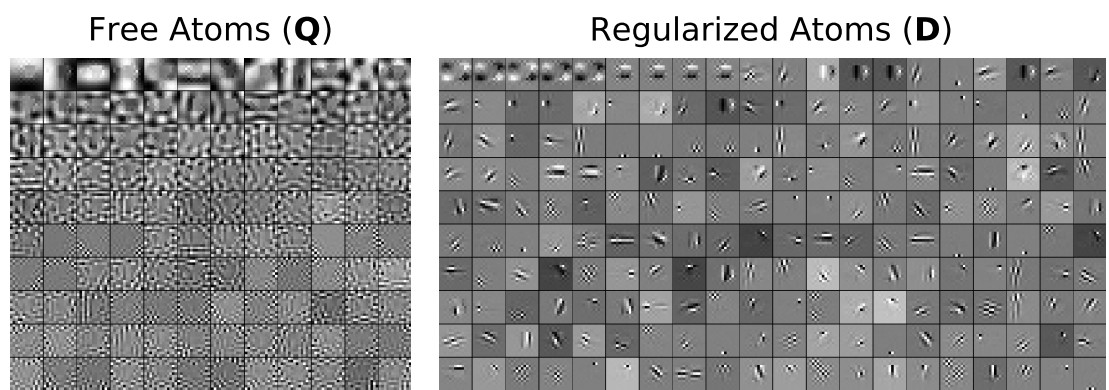

Figure 9: Learned atoms for the NCPR model with $\sigma = 5/255$.

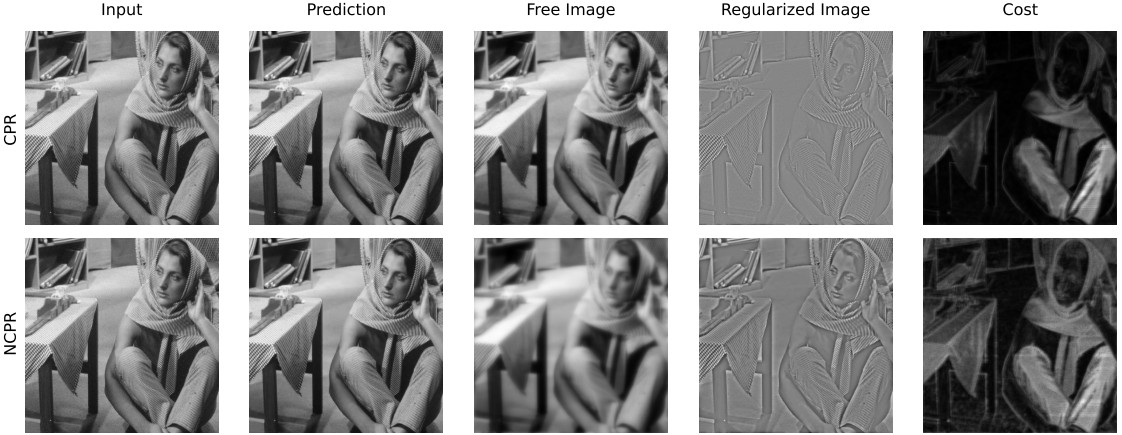

Figure 10: Denoising with $\sigma = 5/255$: decomposition of $\mathbf{x}^*$ into the free-atom and constrained-atom dictionaries. The last column shows the cost associated with each local patch $\mathbf{P}_k\mathbf{x}^*$.

## A.2. Image Reconstructions Visualization

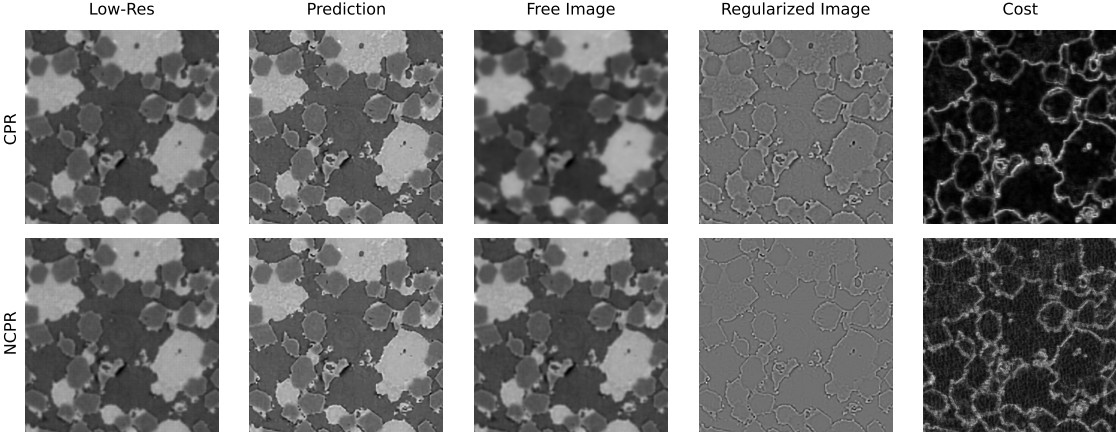

Figure 11: Reconstruction of the super-resolution task: decomposition of $\mathbf{x}^*$ into the free-atom and constrained-atom dictionaries. The last column shows the cost associated with each local patch $\mathbf{P}_k\mathbf{x}^*$.

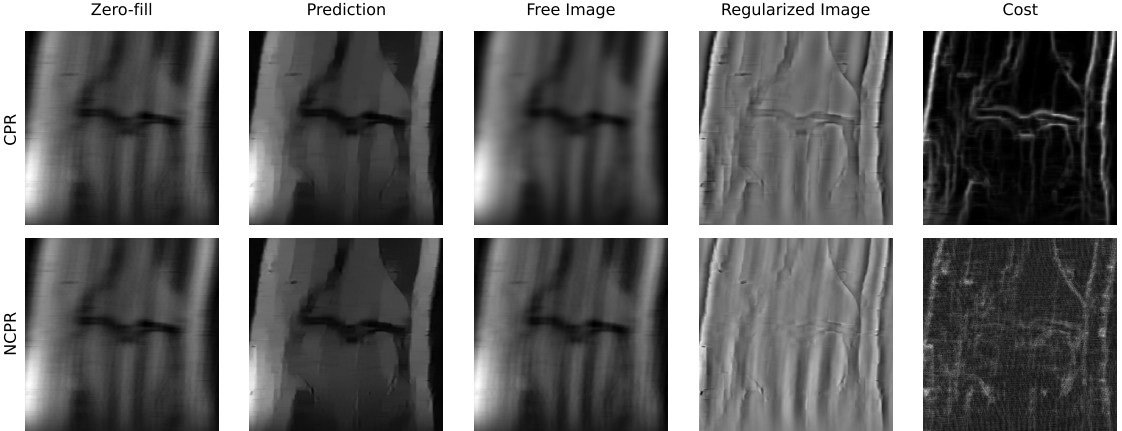

Figure 12: Reconstruction with 8-fold subsampling and PD data: decomposition of $\mathbf{x}^*$ into the free-atom and constrained-atom dictionaries. The last column shows the cost associated with each local patch $\mathbf{P}_k\mathbf{x}^*$.

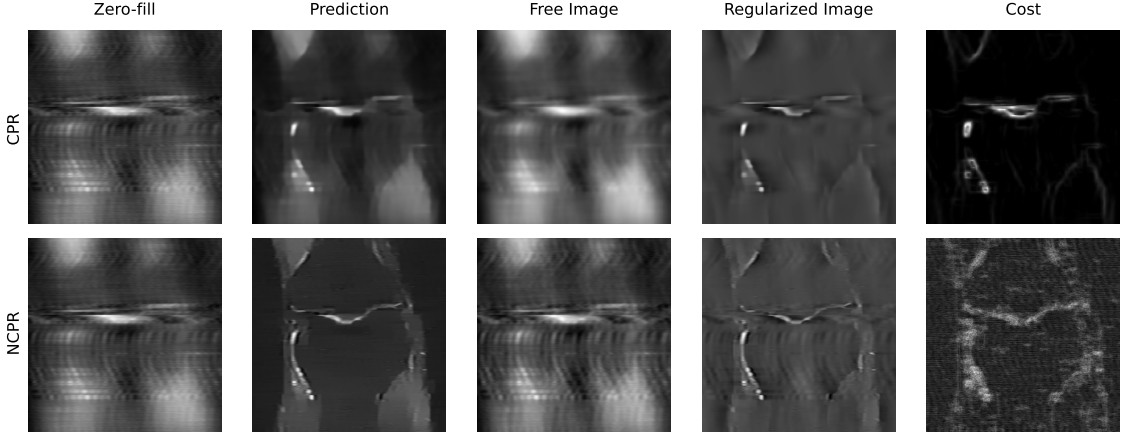

Figure 13: Reconstruction with 16-fold subsampling and PDFS data: decomposition of $\mathbf{x}^*$ into the free-atom and constrained-atom dictionaries. The last column shows the cost associated with each local patch $\mathbf{P}_k\mathbf{x}^*$.

