# OpenReview forum: "Learning of Patch-Based Smooth-Plus-Sparse Models for Image Reconstruction"
_CPAL.cc/2025/Proceedings_Track — CPAL 2025 (Proceedings Track) Poster_

### Official Review · Reviewer_pyUE · 2025-01-06

**Rating:** 5
**Confidence:** 4

**Review:**

**Summary**:  This paper proposes a patch-based method to learn dictionaries and regularizers for linear imaging problems such as super-resolution and CS-MRI.


**Strengths**:
- The paper is very well written and easy to follow.
- The use of two-layer convolutional network to learn the dictionaries and the regularizer.

**Weaknesses/Questions**:
- Why the solution of (3) is indexed by k?
- Missing many recent learning baselines for comparison such MoDL, Deep Image prior, and diffusion-based methods.

**Minor**:
- In MRI, x has complex values not as defined in line 14.
- The acronym of DEQ in line 34 may not be known.

---

### Official Review · Reviewer_e5eU · 2025-01-07
**A patch-based model for image reconstruction**

**Rating:** 7
**Confidence:** 4

**Review:**

This paper introduces a patch-based smooth-plus-sparse model for image reconstruction. The approach combines a penalized sparse representation of image patches with an unconstrained smooth component, yielding interpretable and high-quality reconstructions. The proposed method addresses a bilevel optimization problem, where the inner level uses the traditional algorithms and the outer level leverages supervised learning. Comparisons with conventional and deep learning-based reconstruction methods demonstrate that the proposed framework occasionally surpasses deep learning models in performance.

The paper is well-written, and the experiments are convincing. To further enhance the quality of the manuscript, the authors are requested to address the following points:

1. Please provide detailed information regarding the computational efficiency of the proposed framework and the baseline methods. Specifically, report the training and inference time, as well as the memory requirements, preferably in relation to image resolution. This will help assess the practicality of the proposed methods for realistic high-resolution image reconstruction tasks.

2. It is recommended to include citations to recent patch-based image reconstruction methods, particularly those leveraging deep learning. For your reference

[1] Altekrüger, Fabian, et al. "PatchNR: learning from very few images by patch normalizing flow regularization." Inverse Problems 39.6 (2023): 064006.

[2] Hertrich, Johannes, Antoine Houdard, and Claudia Redenbach. "Wasserstein patch prior for image superresolution." IEEE Transactions on Computational Imaging 8 (2022): 693-704.

[3] Gilton, Davis, Greg Ongie, and Rebecca Willett. "Learned patch-based regularization for inverse problems in imaging." IEEE 8th international workshop on computational advances in multi-sensor adaptive processing (camsap 2019).

[4] Khorashadizadeh, AmirEhsan et al. “LoFi: Neural Local Fields for Scalable Image Reconstruction.” arXiv 2024.

[5] Piening, Moritz, et al. "Learning from small data sets: Patch‐based regularizers in inverse problems for image reconstruction." GAMM‐Mitteilungen (2024): e202470002.

[6] Wang, Zhendong, et al. "Patch diffusion: Faster and more data-efficient training of diffusion models." Advances in neural information processing systems 36 (2024).

---

### Official Review · Reviewer_DWck · 2025-01-10
**Comments on "Learning of Patch-Based Smooth-Plus-Sparse Models for Image Reconstruction"**

**Rating:** 7
**Confidence:** 4

**Review:**

The authors proposed to use automatic differentiation and supervised learning to find the parameters of the regularization term of a variational formulation to solve inverse problems.
The difficulty is that it involves a bilevel optimization problem. The main contribution is to provide an efficient way to solve the inner optimization problem as a fixed-point problem.
Memory complexity caused by using implicit differentiation in the bilevel optimization is bypassed by decomposing the input into patches.
The main interest of this approach is that it relies on strong theoretical results that ensure the stability of the method. This is confirmed by the experimental results of Section 4.

The content of this work is original, connecting standard variational formulation for solving inverse problems, which comes with theoretical guarantees, with more modern prior learned from a dataset. The proposed approach also lies between the two in terms of performance.
The content of this paper is in line with the conference topics.

I have only minor questions, mainly concerning the implementation of the theoretical framework described.
Major comments:
- Can you please comment on the implicit differentiation of the inner solution? Why do you use two different algorithms for different noise level in Section 4.1? Is there a way to select the right one other that trying and checking the results on a validation set?
- I would like to have some clarification on your practical training procedure. Do you use $M=238400$,  $n = m = 40^2$ and $d^2 =  13$? Because in introduction, what you called a patch is an element of size $\sqrt{d} \times \sqrt{d}$, and in Section 4.1 you say that patches are of size $40\times 40$. Can you also specify the batch size if you use any?
- Can you explain why do you choose to compare with TV regularization and K-SVD but not with BM3D or non-local means, which from my understanding, are more commonly used, often refer as state of the art in the non-learning category and also patch-based?
- Can you provide some order of magnitude regarding the time complexity of your method, at least for inference? I would be curious to see how the time and memory complexity of solving (3) evolves with increasing image size or patch size.
- Section 4.2: the visualization is nice. How do you choose $p_1$ and $p_2$ in practice? Is there a way to adjust? After few iterations of training maybe?

Some minor comments:
- Line 108: you introduce the notion of LSI. It might worth expanding it a little, or at least to not use an acronym.
- Results in Table 1: in line 166/167, you mention that your model(s) ' significantly outperform' either TV or TV and the K-SVD. I would not say that an improvement of 0,36dB in PSNR is a 'significant' improvement (which is fine). Could you please nuance your conclusions a little more?
- Section 4.2: The matrix $\mathrm{H}$ is not the same as in Introduction in Equation (1)?

---

### Official Review · Reviewer_pS6W · 2025-01-19
**This paper proposed a patch based dictionary based algorithm for image reconstruction. The method is valid but I have concern over the contribution.**

**Rating:** 7
**Confidence:** 3

**Review:**

1. The paper is well-written, very easy to follow. The method is clearly explained, achieving high performance on solving several inverse problems.

2. The method itself is correct, but I have doubt on the contribution of this work. The key idea (equation 5) is to learn two orthogonal dictionaries $Q$ an $D$ such that they capture the sparse part and the smooth part of the images respectively. If my understanding about this intuition is correct (please correct me if I am wrong), then why not just combine them by searching for a single overcomplete dictionary, which has been widely studied in previous works? I would recommend acceptance of this paper if the authors can better explain the contribution of this work.

---

### Meta-Review · Area_Chair_ta6n · 2025-02-03

**Recommendation:** Accept (Poster)
**Confidence:** 5

**Metareview:**

I find this paper a clear accept. Three of the four reviewers endorse acceptance, highlighting the clarity of the proposed patch-based smooth-plus-sparse image model, solid theory, and strong empirical results on multiple problems (denoising, super-resolution, CS-MRI). The smooth+sparse decomposition is natural and well-motivated (and I'd say classical)—and thus interpretable. The implementation is efficient and yields better results than standard variational and classical dictionary-learning approaches. Comparisons with baselines (TV, K-SVD, and deep-learning models) further strengthen the paper. While one reviewer raised concerns about missing comparisons with additional neural methods (e.g., MoDL or diffusion-based models), the authors clarified that these rely on different training protocols or application scopes outside the present framework. Overall, the consensus is that the paper’s methodological contributions and results merit acceptance.

---

### Decision · Program_Chairs · 2025-02-11

Accept (Poster)